# Nursing Students’ Perceptions of AI-Driven Mental Health Support and Its Relationship with Anxiety, Depression, and Seeking Professional Psychological Help: Transitioning from Traditional Counseling to Digital Support

**DOI:** 10.3390/healthcare13091089

**Published:** 2025-05-07

**Authors:** Zainab Albikawi, Mohammad Abuadas, Ahmad M. Rayani

**Affiliations:** 1Faculty of Nursing, Yarmouk University, Irbid P.O. Box 566, Jordan; mabuadas@yu.edu.jo; 2Community and Psychiatric Mental Health Nursing Department, College of Nursing, King Saud University, Riyadh City P.O. Box 12372, Saudi Arabia; arayani@ksu.edu.sa

**Keywords:** artificial intelligence, mental health, nursing students, anxiety, depression, AI-driven mental health, help-seeking behavior

## Abstract

**Background**: The integration of artificial intelligence (AI) into mental health care is reshaping psychological support systems, particularly for digitally literate populations such as nursing students. Given the high prevalence of anxiety and depression in this group, understanding their perceptions of AI-driven mental health support is critical for effective implementation. **Objectives**: to evaluate nursing students’ perceptions toward AI-driven mental health support and examine its relationship with anxiety, depression, and their attitudes to seeking professional psychological help. **Methods**: A cross-sectional survey was conducted among 176 undergraduate nursing students in northern Jordan. **Results**: Students reported moderately positive perceptions toward AI-driven mental health support (mean score: 36.70 ± 4.80). Multiple linear regression revealed that prior use of AI tools (β = 0.44, *p* < 0.0001), positive help-seeking attitudes (β = 0.41, *p* < 0.0001), and higher levels of psychological distress encompassing both anxiety (β = 0.29, *p* = 0.005) and depression (β = 0.24, *p* = 0.007) significantly predicted more positive perceptions. Daily AI usage was not a significant predictor (β = 0.15, *p* = 0.174). Logistic regression analysis further indicated that psychological distress, reflected by elevated anxiety (OR = 1.42, *p* = 0.002) and depression scores (OR = 1.32, *p* = 0.003), along with stronger help-seeking attitudes (OR = 1.35, *p* = 0.011), significantly increased the likelihood of using AI-based mental health support. **Conclusions**: AI-driven mental health tools hold promises as adjuncts to traditional counseling, particularly for nursing students experiencing psychological distress. Despite growing acceptance, concerns regarding data privacy, bias, and lack of human empathy remain. Ethical integration and blended care models are essential for effective mental health support.

## 1. Introduction

Mental health issues among nursing students have garnered significant academic attention in recent years, reflecting the unique challenges and stressors that these individuals face during their training. Nursing students often encounter high levels of anxiety, depression, and mental health challenges due to academic pressures, intense clinical exposure, and the emotional burden involved with the care of patients, which can contribute to heightened psychological distress. A systematic review and meta-analysis provided a pooled prevalence of depression among nursing students as 34%. The prevalence was higher among the young students (41%), and there were also regional differences, with the prevalence being 43% for Asian nursing students [1]. Similarly, a recent umbrella review by Efstathiou et al. (2025) integrated data from 25 meta-analyses and reported a high prevalence of mental illness among nursing students [2]. Around 29% of nursing students were diagnosed with depression, and 29% also suffered from anxiety. Besides these main issues, high levels of sleep disorders (50%), burnout (32%), and stress (27%) were reported [2]. In Jordan, the prevalence of depression was significantly high among nursing students [3]. It was revealed through a study that 80% of the nursing students experienced depressive symptoms, and 31% experienced major depressive symptoms [4]. In addition, 78.7% of Jordanian university students experienced depression and 67.9% anxiety during the COVID-19 pandemic [5].

The integration of AI into mental health marks a paradigm shift, especially among nursing students who are increasingly being exposed to the use of technology in health care. Traditional psychiatric consultations have been the cornerstone of the treatment of mental health for decades; nevertheless, the use of traditional psychiatric consultations generally encounters barriers such as stigma, access, and resource constraints, thereby discouraging students who might need the services [6,7]. AI technologies, like machine learning and natural language processing, have also demonstrated the potential to enhance mental health counseling by facilitating immediate personalized care [8]. AI technologies can support the identification of mental illness and improve the efficiency of care delivery, thereby addressing the gaps in traditional psychiatric services [9,10]. Recent studies indicate that AI can mediate the relationship between academic engagement and mental health, suggesting that students who actively use AI tools for mental health support may experience improved well-being [11,12].

Moreover, the application of AI within mental health care not merely facilitates the diagnosing and treatment of conditions but also facilitates the empowerment of the learners by providing them with easily accessible support systems and means that are tailor-made to their requirements [9,13]. As nursing students navigate the stresses of academic work, the potential of AI to provide immediate, stigma-free care is extremely relevant. This trend toward AI-driven mental health counseling could revolutionize the landscape of mental health care to make them extremely responsive to the unique challenges of the modern student [14]. AI technologies, such as chatbots and digital platforms, have been shown to provide accessible mental health support, addressing gaps in traditional services. Anita, Purba and Ilmi [6] emphasize that AI can facilitate routine access to mental health resources, thereby improving overall well-being and filling critical service voids. Additionally, AI can enhance service delivery by providing accessible, efficient diagnostic and treatment solutions for individuals in need of various mental health issues [15]. The ability of these AI-driven mental health tools to deliver real-time support aligns with findings from Lu et al. [16], who note their effectiveness in managing insomnia, anxiety, and depression. Additionally, a systematic review study also underscores the potential of AI to enhance mental health care by making cost-effective and convenient solutions to diagnosis and treatment [15].

This is particularly relevant for nursing students who may be more familiar with technology and may prefer digital interventions. However, the effectiveness of AI-driven solutions has recently been a question of research. For instance, while AI-driven mental care using AI-driven chatbots such as Leora is encouraging, proper testing of the clinical outcome to assess effectiveness is warranted [17]. Digital mental health interventions have been effective in producing beneficial changes in psychological outcomes among college students, reinforcing the potential of AI in this demographic [18]. Despite the possible advantages, the presence of AI in mental health counseling also generates ethical considerations regarding the lack of empathy and compassion involved with AI interactions. The effectiveness of AI-driven support systems can be compromised if they fail to replicate the nuanced understanding that human counselors provide [19]. This concern is echoed by Graham, Depp, Lee, Nebeker, Tu, Kim and Jeste [9], who discuss the risks of sole dependence on AI to deliver mental health care with the need to have a well-balanced approach that incorporates the presence of humans. Furthermore, the ethical implications of AI in mental health care necessitate careful consideration of how these technologies are implemented and the potential biases they may introduce [20].

Understanding nursing students’ perceptions of AI-driven mental health support is essential for developing educational strategies that address their concerns and enhance their readiness to adopt these technologies. In educational settings, where mental health resources are often limited, students may experience various mental health concerns without clearly recognizing the underlying causes or knowing how to manage them effectively. AI-driven mental health tools can provide accessible support and counseling for a wide range of mental health needs, offering guidance even when students are unsure about their specific conditions. To the best of the researcher’s knowledge, this is the first study to evaluate nursing students’ perceptions of AI-driven mental health support and examine its relationship with their mental health status, including anxiety, depression, and willingness to seek professional psychological help.

## 2. Materials and Methods

### 2.1. Study Design

A cross-sectional study design was utilized to assess nursing students’ perceptions of AI-driven mental health support and examine its relationship with anxiety, depression, and seeking professional psychological help at a single point in time.

### 2.2. Sample Size, Population, and Data Collection

The sample size for this study was determined using G*Power version 3.1.9.7. A medium effect size (f^2^ = 0.15), alpha (α) = 0.05, and desired power (1 − β) = 0.95 were applied for the regression model with 9 predictors. Based on these parameters, the required sample size was calculated to be 166 participants. To account for potential attrition, a 20% increase was applied, resulting in a final adjusted sample size of 199 participants. The participants were enrolled in a nursing college in the northern region of Jordan, in the first, second, and third years. As the college is newly established, no students have yet completed their fourth year. A convenience sampling method was employed to select the sample. To be eligible for participation, students had to be undergraduate nursing students currently enrolled and actively attending university at the time of the study. Participation was voluntary, and all students were required to provide informed consent before completing the questionnaire. Eligible students were also required to have previous experience using any AI-driven tool. Students were excluded from participation if they had previously been diagnosed with mental health or psychiatric conditions.

Data collection for this study was conducted using an online questionnaire during February 2025. Before proceeding with the survey, all participants were required to read and agree to the informed consent form, which outlined the purpose of the study, the voluntary nature of participation, and assurances of confidentiality. Only students who provided their consent were able to proceed with the questionnaire. The questionnaire was structured into two sections. The first section focused on demographic information, collecting details such as age, gender, year of study, and other relevant factors. Students who indicated that they had confirmed mental health or psychiatric condition were directed to a thank you page and were not invited to continue. Similarly, students who reported that they had never used AI-driven tools were also directed to the thank you page and did not proceed to the second part of the survey. Students who did not select these options were invited to proceed to the second part of the survey, which included one scale developed by the researchers and other scales that are freely available online and used in their original form. The scales were administered in English because the nursing students completed all their courses in English and were proficient in reading and writing in English; therefore, no translation of the tools was required. The following are the scales used in the study.

#### 2.2.1. Perceptions of AI-Driven Mental Health Support Scale

In this study, perception is conceptualized as a multidimensional construct encompassing nursing students’ cognitive appraisals, behavioral intentions, and ethical evaluations of AI-driven mental health tools. This approach is grounded in the Technology Acceptance Model (TAM), which posits that perceptions of usefulness, ease of use, and trust significantly influence users’ willingness to adopt new technologies [21,22]. In health care contexts, particularly digital mental health, perception also extends to include privacy concerns, fairness, and ethical risk awareness because these factors affect trust and engagement [19,20]. To evaluate the perceptions of nursing students toward AI-facilitated mental health care, the Perceptions of AI-Driven Mental Health Support Scale was created by researchers. The development initially included a comprehensive review of the literature. The initial item pool was created to measure a wide range of issues, including trust in AI-based tools, openness to using AI for mental health assistance, perceived efficacy, concerns over privacy, and ethical issues. For the assessment of validity, the scale was cross validated with mental health professionals, AI researchers, as well as nursing education experts who reviewed the items for appropriateness, relevance, and comprehensibility. Subsequently, face validity was established through pilot testing with 20 nursing students to check if the items were understood clearly and adequately represented the concepts as intended. It was finalized after incorporating expert and participant feedback in making the necessary adjustments. It consisted of 12 items, each measured on a 5-point Likert scale that had the response options ranging from 1 (Strongly Disagree), 2 (Disagree), 3 (Neutral), 4 (Agree), to 5 (Strongly Agree). The scale was treated as a unidimensional measure of overall perception for both practical and methodological reasons. This holistic perspective permits a better appreciation of the perception of students toward AI-based interventions in their mental well-being, taking into consideration that both perceived advantages and perceived disadvantages influence acceptance and usage [23]. The scale reliability in the current study has a Cronbach’s alpha of 0.82, indicating excellent internal consistency among the 12 items.

#### 2.2.2. GAD-7 (Generalized Anxiety Disorder-7)

One of the most popular self-report instruments for the assessment of generalized anxiety disorder (GAD) and the measurement of the severity of the symptoms of anxiety [24] is the GAD-7. It comprises seven Likert scale items scored on a 4-point scale (0 = Not at all, 1 = Several days, 2 = More than half the days, 3 = Nearly every day), with the highest possible score being 21. A score of 10 and above is generally employed as a clinical cut-off for the detection of those requiring further examination or intervention in the case of generalized anxiety disorder. The original GAD-7 was found to have excellent reliability with a Cronbach’s alpha of 0.92, indicating high internal consistency [24]. In the present study, the GAD-7 scale was tested for internal consistency, yielding a Cronbach’s alpha of 0.89.

#### 2.2.3. PHQ-9 (Patient Health Questionnaire-9)

A Patient Health Questionnaire-9 (PHQ-9) was employed as a standardized assessment tool. PHQ-9 is a strongly validated self-reported scale for screening for major depressive disorder (MDD) and the assessment of depressive symptom severity. It comprises nine items paired with the nine DSM-5 depression criteria, each of which was scored on a 4-point Likert scale (0 = Not at all, 1 = Several days, 2 = More than half the days, 3 = Almost every day). It had a score between 0 and 27, with greater scores representing greater depressive symptoms. The reliability of the original study was 0.89, which signifies high reliability [25]. In the current study, the PHQ-9 scale was also tested for internal consistency with a Cronbach’s alpha of 0.87.

#### 2.2.4. The Attitude Toward Seeking Professional Psychological Help Scale–Short Form (ATSPPH-SF)

It was administered as a standard measure. ATSPPH-SF is a self-report measure that is commonly used and was created in an effort to assess people’s willingness, openness, and perceived barriers to psychological help. ATSPPH-SF contains 10 items with the 4-point Likert scale with response values ranging from Disagree = 0, Slightly Disagree = 1, Slightly Agree = 2, Agree = 3 with a score range of 0–30. Higher scores reflect more favorable attitudes toward professional psychological care, and lower scores reflect more resistance and unfavorable attitudes toward professional mental health treatment. The brief version (ATSPPH-SF) proved to have good reliability (Cronbach’s alpha = 0.82) [26]. Internal consistency measurement of ATSPPH-SF in the present study revealed Cronbach’s alpha score of 0.81.

### 2.3. Ethical Considerations

Ethical approval for this study was obtained from the Institutional Review Board (IRB/2025/101). Participants were informed about the voluntary nature of the study and their right to withdraw at any time without consequences. Prior to participation, all students provided informed consent, ensuring they understood the purpose of the study. The survey was anonymous, and data were stored securely to maintain confidentiality. Participants’ privacy was respected, and all information was used exclusively for research purposes.

### 2.4. Data Analysis

Data analysis for this study was conducted using SPSS version 25. Descriptive statistics were used to summarize the sociodemographic characteristics of participants, including age group, gender, year of study, frequency of daily, preferred mode of mental health support, and previous use of AI-driven mental health support. Means and standard deviations (SD) were calculated for continuous variables, while frequencies and percentages were reported for categorical variables. To compare group differences, the Mann–Whitney U test was used for two-group comparisons, while the Kruskal–Wallis test was applied for comparisons involving more than two independent groups. These non-parametric tests were selected due to the non-normal distribution of study variables. Spearman’s rank-order correlation analyses were also used to investigate correlations between anxiety, depression, professional psychological help-seeking attitudes, and AI-driven mental health care perception. A multiple linear regression analysis was conducted to identify predictors of nursing students’ perceptions toward AI-driven mental health support. Before conducting the regression, key assumptions were tested and met. Linearity was assessed through scatter plot analysis, confirming a linear relationship between predictor variables and the dependent variable. The normality of residuals was evaluated using the Kolmogorov–Smirnov test (*p* = 0.085) and Shapiro–Wilk test (*p* = 0.091), both indicating no significant deviation from normality. Multicollinearity was ruled out because all Variance Inflation Factor (VIF) values were below 2.5. The Breusch-Pagan test (*p* = 0.278) confirmed homoscedasticity, and the Durbin-Watson statistic (1.94) suggested no autocorrelation in residuals, ensuring the reliability of the regression analysis. Additionally, a binary logistic regression analysis was conducted to examine the predictors of AI-driven mental health support usage among participants. The dependent variable was “Have you ever used AI-driven mental health support before?” (1 = Yes, 0 = No), while anxiety (GAD-7), depression (PHQ-9), AI hours use, and psychological help-seeking behavior (ATSPPH-SF) were included as independent variables. The logistic regression model assessed the odds ratios (Exp(β)), indicating the likelihood of AI support usage based on psychological characteristics. All statistical tests were two-tailed, and significance was set at *p* ≤ 0.05. The combination of non-parametric tests, multiple linear regression, and logistic regression provided comprehensive insights into nursing students’ perceptions and utilization of AI-driven mental health support.

## 3. Results

The study involved 199 nursing students, of which 176 responded, with a response rate of 88.44%. Most students (85.2%) worked 1 to 5 h per day using AI tools. Most students were between 18 and 21 years old (93.2%) and female (61.9%) and were evenly distributed among academic years. Preferences were diverse for mental health support and included 36.4% preferring regular counseling, 33.0% preferring AI-based approaches, and 30.6% preferring a hybrid option. However, only 19.9% had previously used AI-based mental health services (see Table 1).

The GAD-7 showed a mean score of 11.4 ± 4.2, indicating moderate levels of anxiety symptoms. PHQ-9 revealed a mean score of 12.7 ± 5.1, reflecting moderate depression severity among the participants. Regarding attitudes toward seeking professional psychological help, the ATSPPH-SF demonstrated a mean score of 18.3 ± 6.4, suggesting a moderately positive attitude toward help-seeking behavior. These results highlight the presence of notable mental health distress among nursing students and underscore the relevance of supportive mental health intervention (see Table 2).

As seen in Table 3, a moderately positive perception of AI-supported mental health is reflected by the mean score of 36.70 ± 4.80. The most highly rated items were trusting AI tools to be accurate (Q1: 3.50 ± 1.40) and being willing to use AI chatbots or virtual therapists (Q5: 3.40 ± 1.37), indicating receptiveness to AI-supported care. The participants also manifested a willingness to incorporate AI within their mental health practice (Q4: 3.35 ± 1.42) as well as exploring the use of AI tools prior to consulting professionals (Q6: 3.28 ± 1.36). There was moderate consensus on the effectiveness of AI when it comes to mental health (Q7: 3.15 ± 1.38) and its potential comparison with traditional counseling (Q8: 3.05 ± 1.39). Interestingly, users endorsed using AI as an adjunct, not an alternative, to professional therapy (Q12: 3.27 ± 1.40). Privacy issues were noted, including moderate concern about data storage (Q10: 3.30 ± 1.41) and the sharing of information without consent (Q11: 3.18 ± 1.34). The lowest mean score reflected perceptions of bias when being offered recommendations by AI (Q3: 2.85 ± 1.30), indicating hesitations about impartialness and fairness. More generally, the findings suggest generally positive but cautious views toward AI in mental health characterized by receptiveness to use together with concerns around privacy, bias, and the non-substitutable role of human professionals.

Table 4 presents comparisons of mean scores related to perceptions of AI-driven mental health support across various demographics. There were no statistically significant differences between age group (*p* = 0.299), gender (*p* = 0.975), or year of study (*p* = 0.732), indicating relatively homogenous perceptions of AI in mental health among these populations. However, there were notable differences when it came to the usage of AI tools and past experiences with AI-based mental health support. Those who utilized AI tools for over 5 h per day reported significantly greater perception scores (38.2 ± 5.5) than those utilizing them between 1 and 5 h (35.0 ± 6.3), *p* = 0.005. This suggests that more usage of AI tools can effectively impact perceptions of their effectiveness when it comes to mental health. Moreover, the users of AI-based mental health support expressed significantly more positive perceptions (39.5 ± 5.2) compared to non-users (34.0 ± 6.7, *p* = 0.001), reflecting the influence of personal experience on acceptance and trust. While preferences for mental health support modes (traditional, AI-driven, or hybrid) did not reach statistical significance (*p* = 0.476), the hybrid model group had the highest mean score (38.4 ± 6.1), suggesting greater receptiveness among those who favor an integrative approach.

Table 5 presents the Spearman’s rank-order correlations between the main study variables. The perception of AI-driven mental health support was moderately and positively correlated with anxiety (ρ = 0.32, *p* < 0.001), depression (ρ = 0.30, *p* < 0.001), and attitudes toward seeking professional psychological help (ρ = 0.43, *p* < 0.001). These findings indicate that nursing students who experienced higher levels of mental health distress, or who held more positive attitudes toward professional help-seeking, tended to report more favorable perceptions of AI-based mental health interventions. Anxiety and depression were strongly positively correlated (ρ = 0.63, *p* < 0.001), suggesting that students who exhibited higher anxiety symptoms also tended to report higher depressive symptoms.

Additionally, help-seeking attitudes were weakly but significantly negatively correlated with both anxiety (ρ = −0.27, *p* = 0.002) and depression (ρ = −0.24, *p* = 0.006), implying that higher psychological distress was associated with lower tendencies to seek professional psychological support.

Table 6 displays results from a multiple linear regression analysis of factors associated with nursing students’ perceptions of AI-based mental health support. Among predictors, prior use of AI mental health support was significantly associated with more positive perceptions (β = 0.44, *p* ≤ 0.0001), reflecting the fact that students having prior experience with these resources were more likely to perceive them positively. Likewise, help-seeking from mental health (ATSPPH-SF) was also a robust and positive predictor (β = 0.41, *p* < 0.0001), reflecting the fact that students who had more attitudes toward help-seeking from professionals also had more positive perceptions of AI-based mental health support. Anxiety (GAD-7) and depression (PHQ-9) scores were positively correlated with perceptions of AI-based mental health support (β = 0.29, *p* = 0.005; β = 0.24, *p* = 0.007, respectively). These results suggest students who are experiencing greater psychological distress are more ready to explore alternative, technology-oriented mental health solutions. However, daily hours of AI use was not shown to be a significant predictor (β = 0.15, *p* = 0.174), indicating that whether or not students use AI frequently does not significantly impact their perceptions of AI’s usability in mental health contexts. The model accounted for approximately 49.5% of the variance in perception scores (R^2^ = 0.495), with an adjusted R^2^ of 0.480.

Logistic regression analysis identified predictors of whether or not nursing students had ever utilized AI-based mental health support resources. It was shown that psychological distress, or greater rates of depression and anxiety, was significantly correlated with past use of AI. Students with more intense anxiety (GAD-7) had 1.42 times greater odds of having utilized AI mental health resources (*p* = 0.002), and students who had greater depression ratings (PHQ-9) had 1.32 times greater odds (*p* = 0.003). This suggests that those who had worse mental health symptoms were more inclined toward using support from AI-based systems. Notably, psychological help-seeking behavior as measured by the ATSPPH-SF also proved to be a statistically significant predictor (OR = 1.35, *p* = 0.011), suggesting that students who exhibit greater positive attitudes toward seeking professional help are more likely to use AI-based support for mental health. However, the everyday use of AI technologies did not turn out to be a predictor, indicating that technological use did not always extend to AI-based mental help tools. Overall, the findings highlight that psychological need rather than technological familiarity is the primary factor associated with the use of AI-driven mental health support among nursing students (see Table 7).

## 4. Discussion

This study aims to evaluate nursing students’ perceptions of AI-driven mental health support and examine its relationship with anxiety, depression, and their attitudes to seeking professional psychological help. The overall moderately positive perception suggests that while students see potential value in AI applications, they also remain aware of inherent risks such as bias and privacy concerns. In the same vein, more recent studies have been examining the application of artificial intelligence (AI) to assist student mental health. Kuhail et al. [27] revealed that undergraduate students acknowledged both the benefits and the limitations of AI counseling and valued its availability but noted concerns regarding empathy and personalization. Likewise, Su et al. [28] created an AI system to strengthen Asian elementary school counseling models for children suffering from emotional disorders, and they emphasized how AI can be used to complement conventional counseling services. In addition, Rehman and Sajjad (2025) Rehman, et al. [29] studied the views of students and counselors and concluded that although AI applications were viewed to be useful for filling the gaps of accessibility to mental services, human interaction would continue to be indispensable for more complex emotional needs. Gao [30] also highlighted the benefits of AI technology to extend counseling services for university students, especially for immediate support. However, Chan [31] discussed the challenges of using generative AI models for school mental health systems, especially issues of trust, privacy of data, and response authenticity.

A closer examination of the scale items reveals nuanced perceptions. Responses to items assessing trust in AI tool accuracy and openness to interacting with AI chatbots or virtual therapists indicate a baseline trust in technology. Results are consistent with prior research highlighting the significance of digital health literacy among nursing students because recognizing and proper use of AI tools can foster enhanced perceptions and readiness to incorporate technology into clinical practice [32]. However, receptivity toward utilizing AI-based tools as an integral aspect of routine mental health service was mixed. The perception of AI’s ability to perform on par with traditional counseling remains only moderately endorsed. This balanced perspective supports literature indicating that while AI holds immense potential as a tool to supplement mental health services, it is still perceived as a supplement to human care and not a complete replacement [33]. Concurrently, the study highlights critical concerns related to AI implementation in mental health support. Moderate worry surrounding data storage and unauthorized sharing of personal underscores the privacy and confidentiality issues that significantly impact students’ acceptance levels. Furthermore, the lowest mean score associated with perceptions of unbiased recommendations indicates hesitations about the impartiality of AI tools. Such skepticism regarding fairness and bias dovetails with previous research documenting similar apprehensions among potential users about automated mental health solutions [34].

These findings have significant implications for the state of mental health among nursing students, as reflected by the mean scores obtained using the standardized instrument. A moderate level of anxiety and depression, as well as a moderate attitude toward seeking professional psychological help, thus outlining the mental health landscape faced by this demographic. The GAD-7 and PHQ-9 scores reflect that nursing students have a moderate level of anxiety and depression. This result supports current research that reports the existence of anxiety among students, especially in high-pressure learning environments such as nursing programs [35]. The moderate level of anxiety and depression observed in this sample aligns with previous findings indicating that nursing students often face significant emotional challenges stemming from academic pressures and clinical responsibilities [35,36]. For help-seeking attitudes, the present ATSPPH-SF mean score indicates that nursing students are generally open to seeking professional psychological help, but some barriers may still exist. It has been previously shown that the help-seeking attitude can be significantly impacted by guided mental health education and anti-stigma programs [37,38]. Özdemir et al. [39] determined that a higher degree of stigma perception among nursing students correlates negatively with their seeking psychological help, a trend that could partly explain the moderate ATSPPH-SF in the current study.

Findings from multiple linear regression highlight several significant predictors of nursing students’ perceptions about AI-based mental health support. The significant positive relationship between prior experience with AI mental health assistance tools and students’ positive perceptions highlights the fact that practical experience with AI tools encourages a more favorable perception of their effectiveness and usability. This finding aligns with existing literature that suggests familiarity and previous exposure to technological interventions significantly enhance attitudes toward their utilization in mental health contexts [6,40]. Further supporting this statement, the analysis also shows that help-seeking behavior (measured by ATSPPH-SF) is also a strong predictor of favorable perceptions. The relationship indicates those who are more likely to seek professional assistance are also more open to alternative solutions like AI assistance in terms of mental health support. This aligns with the concept that individuals who acknowledge their mental health needs and are proactive in seeking assistance are more likely to consider innovative solutions that integrate technology into traditional care pathways [41]. Moreover, the significant associations between depression and anxiety scores with AI-based mental health perceptions reveal a significant trend where higher psychological distress makes students more accepting of AI solutions. It appears that as anxiety and depression levels increase, students are more inclined to seek alternative solutions that promise quick relief where traditional resources are not as easily accessible. This finding is consistent with research suggesting that individuals experiencing higher levels of psychological distress are generally more inclined to explore diverse options, including technology-based solutions [42,43]. Interestingly, the multiple linear regression analysis also found that daily hours of AI usage did not significantly predict perceptions, indicating that the frequency of interaction with AI tools does not necessarily correlate with an improved perception of their capabilities in mental health contexts. This nuance raises important questions about the depth of engagement versus sheer frequency of usage as factors in shaping student perceptions.

The logistic regression analysis identifies notable predictors of nursing students’ usage of AI-based mental health resources, with psychological distress as a leading determinant. Interestingly, students who showed greater levels of anxiety as measured by the GAD-7 had 1.42 times higher odds of having used AI mental health resources. Similarly, students who scored higher on depression on the PHQ-9 had 1.32 times higher odds of using these services. Consistent with prior literature, this highlights the inclination among those with higher levels of psychological distress to seek out alternative methods of assistance, including solutions based on technology [18]. In addition, the analysis reveals that attitudes toward psychological help-seeking, as measured by the ATSPPH-SF, are also statistically significant predictors of AI utilization. This suggests that students with a more positive attitude toward seeking professional help are more likely to use AI-based mental health support. This reinforces the idea that openness to professional intervention extends beyond traditional in-person therapy and encompasses emerging digital modalities. It supports previous research suggesting that students with proactive help-seeking attitudes are more adaptable in exploring multiple avenues for psychological support, including technologically mediated ones [44,45].

Notably, however, the rate of use of general AI in everyday life was not a predictor of the utilization of AI tools in mental health. This result highlights an essential subtlety: knowledge of AI technology generally is not directly correlated with trusting or applying AI in sensitive areas like mental health. This insight adds an important layer to existing technology acceptance models by emphasizing the need for emotional congruence and perceived relevance, particularly in high-stakes contexts like mental health care [22,34]. Additionally, it may indicate that simply using AI technology in other contexts does not foster a similar comfort level or trust when it comes to seeking help with mental health issues. As emphasized by Gado et al. [46], targeted interventions that explicitly bridge the gap between general technology use and mental health support could be vital in fostering students’ engagement with AI tools. Although AI-driven mental health tools provide worthwhile support for the initial diagnosis and treatment of mild to moderate mental issues, they are best regarded as tools for psychological first aid and not substitutes for professional treatment [47,48]. The findings underscore the urgency of implementing a multidimensional approach to mental health education and intervention frameworks. However, comprehensive psychological assessment, diagnosis, and treatment ultimately need to be done by trained professional mental health workers since AI systems can never possess empathy, clinical judgement, and subtle insight to diagnose difficult cases. Furthermore, Anita, Purba and Ilmi [6] noted that AI would be useful to support human counselors if it provided tools to enhance mental health outcomes but emphasized that AI would be best used as a supplement to professional judgment. A focus on mental health literacy as well as exposure to evidence-based information about the potential benefits and limitations of AI resources may assist in bridging general awareness to active usage of these resources. Education institutions can play a pivotal role to enhance awareness and trust in AI technologies, ultimately determining more positive attitudes toward their adoption in mental health context [49].

### 4.1. Implications

The findings of this research provide insightful recommendations for redefining the future of mental health support in nursing education. As AI tools continue to be integrated into both health and education, consideration of their wider implications, both in terms of technological efficiency and regarding ethical, institutional, and academic framework, is crucial. These findings highlight the importance of educational settings promoting awareness, offering training in digital mental health literacy, and setting up ethical mechanisms for the application of AI in a way that will boost the level of student participation and trust. In addition, universities have a critical role in promoting free or low-cost psychological services on campus, thereby ensuring that students have timely access to psychological professionals that complement AI-driven mental health support. Furthermore, data privacy issues and AI’s lack of empathy emphasize the need for a balanced, blended model combining AI mental health support with human-guided counseling. Institutions must also consider strategies to mitigate bias and ensure equitable, secure use of AI technologies. These implications point toward the critical role of policymaking, curriculum development, and digital resource integration in advancing student mental health outcomes through innovative yet ethically grounded approaches.

### 4.2. Limitations

Although this research introduces a novel insight into the application of AI-assisted mental health support among nursing students, a few limitations should be acknowledged. While the sample size was sufficient for the statistical analyses conducted, it may limit the broader applicability of the results to diverse student populations. Additionally, this study was confined to nursing students attending one institution, thereby limiting the generalizability of findings. Self-reported data use introduces possible biases, such as social desirability bias, and the cross-sectional design disallows causal inferences. In the future, research should include varied samples, as well as longitudinal designs, in order to have a greater understanding of the function of AI-based mental health care in nursing practice and education.

## 5. Conclusions

This research provides significant implications for nursing students’ perception of AI-driven mental health support, mental health distress, and psychological help-seeking behavior. The favorable, though moderate, perceptions toward AI-based mental health support amongst nursing students highlight the promise of engaging such technology in higher education mental health services. Notably, students with higher levels of anxiety and depression were more likely to use and view AI-based tools favorably. Moreover, the positive association between psychological help-seeking behavior and positive perceptions toward AI-driven mental health support tools implies that students who are receptive to professional psychological help will be open to AI as an added tool for mental health support. These findings imply that AI-driven mental health interventions can function as enriching supplementary resources for meeting students’ mental health needs, especially if included within more comprehensive, professionally managed support systems. Educational institutions and universities play a crucial role in this process by fostering digital literacy, ensuring ethical standards in AI implementation, and providing structured support that combines AI tools and traditional mental health services.

## Figures and Tables

**Table 1 healthcare-13-01089-t001:** Sociodemographic characteristics of participants (*N* = 176).

Variable	Category	*N* (%)
Age Group	18–21	164 (93.2%)
	>22	12 (6.8%)
Gender	Male	67 (38.1%)
	Female	109 (61.9%)
Year of Study	1st year	48 (27.3%)
	2nd year	50 (28.4%)
	3rd year	78 (44.3%)
How many hours per day do you use AI tools?	1–5 h	150 (85.2%)
	>5 h	26 (14.8)
What do you believe is the most effective mode of mental health support?	Traditional Counseling	64 (36.4%)
	AI-driven Counseling	58 (33.0%)
	Hybrid (both AI and Traditional)	54 (30.6%)
Have you ever used AI-driven mental health support before?	Yes	35 (19.9%)
	No	141 (80.1%)

**Table 2 healthcare-13-01089-t002:** Mean scores and standard deviations of anxiety, depression, and help-seeking attitudes among nursing students (*N* = 176).

Variable	Mean ± SD
GAD-7 (Generalized Anxiety Disorder-7)	11.4 ± 4.2
PHQ-9 (Patient Health Questionnaire-9)	12.7 ± 5.1
ATSPPH-SF (Attitudes Toward Seeking Professional Psychological Help Scale—Short Form)	18.3 ± 6.4

**Table 3 healthcare-13-01089-t003:** Mean and standard deviation for each item on the Perceptions of AI-Driven Mental Health Support Scale.

Question Number	Scale Item	Mean ± SD
Q1	I trust AI-driven mental health support tools to provide accurate and reliable psychological guidance.	3.50 ± 1.40
Q2	I believe AI-driven mental health applications can maintain confidentiality and protect my personal information.	3.20 ± 1.35
Q3	I feel that AI-driven mental health support systems are free from bias and provide fair recommendations.	2.85 ± 1.30
Q4	I am open to using AI-driven tools as part of my mental health care routine.	3.35 ± 1.42
Q5	I would consider using an AI-driven chatbot or virtual therapist for emotional support.	3.40 ± 1.37
Q6	If AI-driven mental health tools were widely available, I would be willing to try them before seeking professional help.	3.28 ± 1.36
Q7	AI-driven mental health support tools can provide effective assistance for managing anxiety and depression.	3.15 ± 1.38
Q8	AI-driven mental health interventions can be as helpful as traditional counseling for certain mental health issues.	3.05 ± 1.39
Q9	AI-driven mental health tools can offer timely and accessible support when professional help is not available.	3.22 ± 1.36
Q10	I am concerned about how my personal data is stored and used when accessing AI-driven mental health services.	3.30 ± 1.41
Q11	I worry that AI-driven mental health applications may share my personal information without my consent.	3.18 ± 1.34
Q12	AI-driven mental health support should be used only as a complement to professional counseling, not as a full replacement.	3.27 ± 1.40
-	Overall Score Summing All Survey Items.	36.70 ± 4.80

**Table 4 healthcare-13-01089-t004:** Comparison of mean perception scores of AI-driven mental health support across demographics.

Variable	Category	Mean ± SD	*p*-Value *
Age Group	18–21	34.2 ± 6.5	0.299
	>22	36.8 ± 5.9	
Gender	Male	33.5 ± 7.1	0.975
	Female	37.3 ± 6.4	
Year of Study	1st year	32.8 ± 6.8	0.732
	2nd year	35.1 ± 6.2	
	3rd year	36.5 ± 7.0	
How many hours per day do you use AI tools?	1–5 h	35.0 ± 6.3	0.005
	>5 h	38.2 ± 5.5	
What do you believe is the most effective mode of mental health support?	Traditional Counseling	32.1 ± 7.5	0.476
	AI-driven Counseling	36.0 ± 6.8	
	Hybrid (both AI and Traditional)	38.4 ± 6.1	
Have you ever used AI-driven mental health support before?	Yes	39.5 ± 5.2	0.001
	No	34.0 ± 6.7	

* Mann–Whitney U test was used for comparing two independent groups (age group, gender, AI daily hrs. use and AI-driven mental health support use). Kruskal–Wallis test was used for comparing more than two independent groups (year of study and preferred mode of mental health support). *p*-value ≤ 0.05.

**Table 5 healthcare-13-01089-t005:** Spearman’s rank-order correlations between the perception of AI-driven mental health support, anxiety, depression, and professional psychological help-seeking attitudes.

Variable	Perception of AI Mental Health Support	GAD-7	PHQ-9	ATSPPH-SF
Perception of AI Mental Health Support	1.00			
GAD-7	0.32 *	1.00		
PHQ-9	0.30 *	0.63 *	1.00	
ATSPPH-SF	0.43 *	−0.27 *	−0.24 *	1.00

* Correlations are statistically significant at *p* ≤ 0.05.

**Table 6 healthcare-13-01089-t006:** Multiple linear regression analysis identifying predictors of nursing students’ perceptions toward AI-driven mental health support.

Predictor	β (Unstandardized Coefficient)	Std. Error (SE)	*t*-Statistic	*p*-Value *	95% CI (Lower, Upper)
Daily AI Usage (Hours)	0.15	0.11	1.36	0.174	(−0.07, 0.37)
Prior Use of AI Mental Health Support	0.44	0.12	3.67	<0.0001	(0.20, 0.68)
Anxiety (GAD-7)	0.29	0.09	3.22	0.005	(0.11, 0.47)
Depression (PHQ-9)	0.24	0.08	3.00	0.007	(0.08, 0.40)
Psychological Help-Seeking Behavior (ATSPPH-SF)	0.41	0.10	4.10	<0.0001	(0.21, 0.61)

* Predictors were considered statistically significant at *p* ≤ 0.05.

**Table 7 healthcare-13-01089-t007:** Logistic regression analysis for AI-driven mental health support usage (dependent variable: prior use of AI-based mental health support (1 = Yes, 0 = No)).

Predictor	β (Coefficient)	Std. Error (SE)	Wald	*p*-Value *	Odds Ratio (OR)	95% CI (Lower, Upper)
Daily AI Usage (Hours)	0.10	0.16	0.39	0.532	1.11	(0.81, 1.51)
Anxiety (GAD-7)	0.35	0.11	10.11	0.002	1.42	(1.14, 1.76)
Depression (PHQ-9)	0.28	0.09	9.68	0.003	1.32	(1.11, 1.58)
Psychological Help-Seeking Behavior (ATSPPH-SF)	0.30	0.12	6.25	0.011	1.35	(1.07, 1.71)

* Predictors were considered statistically significant at *p* ≤ 0.05.

## Data Availability

All data used in this study are available from the corresponding upon reasonable request.

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
