# Peer review of "Nursing Students’ Perceptions of AI-Driven Mental Health Support and Its Relationship with Anxiety, Depression, and Seeking Professional Psychological Help: Transitioning from Traditional Counseling to Digital Support"

_healthcare, 2025, doi:10.3390/healthcare13091089_

Round 1

Reviewer 1 Report

Comments and Suggestions for Authors

Dear Authors,

Congratulations on your intriguing and timely paper. The research linking mental health issues among students with the use of artificial intelligence in this context represents an important contribution to the current scientific and societal discussion.

However, I believe the introduction could be structured more clearly. I recommend that the core issue—deteriorating mental health among students, particularly nursing students, along with a brief overview of the prevalence of anxiety and depression—be precisely defined right at the beginning. Following this, artificial intelligence should be introduced as a potential solution that can help alleviate barriers such as stigma and limited access to mental health services.

Additionally, it would be helpful to more clearly highlight existing findings in the literature, as well as identify the gaps that this study aims to address. At the end of the introduction, I suggest clearly stating the research aim to provide readers with better guidance throughout the rest of the paper.

On the other hand, I would like to especially commend the results and discussion sections. The findings are clearly presented and interpreted based on relevant literature, and the discussion thoughtfully balances the potential and limitations of AI in the context of mental health. It is particularly valuable that the ethical dimensions, as well as the role of educational institutions in the further implementation of these tools, are highlighted.

Best regards

Comments on the Quality of English Language

I recommend that the entire text be thoroughly reviewed as there are several writing errors present, such as the improper formatting of reference numbers, which are not separated from the text. These minor issues can disrupt the flow and clarity of the paper. A careful revision will enhance the overall readability and professionalism of the manuscript.

Author Response

We sincerely thank you for your thoughtful and constructive review of our manuscript. Your comments have significantly improved the quality and clarity of the work. We have carefully addressed each comment, and the corresponding revisions have been highlighted in red within the revised manuscript. Your insights were instrumental in improving the overall rigor and relevance of the study.

Comment 1: Congratulations on your intriguing and timely paper. The research linking mental health issues among students with the use of artificial intelligence in this context represents an important contribution to the current scientific and societal discussion.

Response 1: We appreciate your positive feedback. We are pleased that our work contributes meaningfully to the ongoing scientific discourse on mental health and artificial intelligence.

Comment 2: However, I believe the introduction could be structured more clearly. I recommend that the core issue—deteriorating mental health among students, particularly nursing students, along with a brief overview of the prevalence of anxiety and depression—be precisely defined right at the beginning. Following this, artificial intelligence should be introduced as a potential solution that can help alleviate barriers such as stigma and limited access to mental health services.

Response 2: Thank you for pointing this out. We agree with this comment. Therefore , we have revised the introduction to first highlight the mental health challenges faced by nursing students, followed by the prevalence of anxiety and depression Subsequently, we introduce artificial intelligence as a potential solution. Pages 1-3.

Comment 3: Additionally, it would be helpful to more clearly highlight existing findings in the literature, as well as identify the gaps that this study aims to address. At the end of the introduction, I suggest clearly stating the research aim to provide readers with better guidance throughout the rest of the paper.

Response 3 : Thank you for pointing this out. We agree with this comment. Therefore , we have incorporated the available literature on students’ use of AI-driven tools for mental health support, acknowledging that this area remains underexplored. We have clearly identified the gap in the existing research and revised the introduction to include the aim of the study. Page 3, line 100-110.

Comment 4: On the other hand, I would like to especially commend the results and discussion sections. The findings are clearly presented and interpreted based on relevant literature, and the discussion thoughtfully balances the potential and limitations of AI in the context of mental health. It is particularly valuable that the ethical dimensions, as well as the role of educational institutions in the further implementation of these tools, are highlighted.

Response 4: We greatly appreciate your recognition of the clarity and balance in the results and discussion sections. We have also further emphasized the ethical dimensions and the role of educational institutions by adding specific sentences to the Implications section to highlight their importance in the responsible implementation of AI-driven mental health tools. This can be found on page 14, specifically lines 500-515.

Comment 5: I recommend that the entire text be thoroughly reviewed as there are several writing errors present, such as the improper formatting of reference numbers, which are not separated from the text. These minor issues can disrupt the flow and clarity of the paper. A careful revision will enhance the overall readability and professionalism of the manuscript.

Response 5: Thank you for pointing out the formatting issues. We used EndNote to manage references and citations, which may have caused the text's improper spacing and formatting. We have carefully reviewed the entire manuscript and made the necessary corrections to ensure clarity, consistency, and improved readability.

Reviewer 2 Report

Comments and Suggestions for Authors Introduction: This section needs more focus on the studied population. Also, the references which are cited do not match the text where they are cited - this must be thoroughly checked. For example, on Lines 47-50: The cited references No. 9 and 10 have nothing to do with the text of this sentence - ref. 9 is a preprint (there is an article published) which describes the differences in the "load" of psychiatric consultations before and following the onset of pandemic and ref. No. 10 is a case report of a woman, describing the impact of pandemic on her mental health, while the text of the sentence is about the adoption and effectiveness of AI and telemedicine. This must be corrected and checked throughout the paper.  Line 93: "As a result.." - results of what? How does this sentence arise from the entire paragraph? Lines 97-98: The cited tool refers to which mental health issue experienced by students? It should be stated precisely.  Lines 103-114: This paragraph lacks coherency. It should summarize the main points and outline the aims of the work. For example, what does the text on Line 107 mean - what resources are limited especially in the educational settings. Line 133: Does any AI-driven tool mean not only related to mental health but any at all? Methods: State explicitly whether approval for the use of questionnaires was received from their owners, or if not required - state so. Methods: State the time when the study was conducted. Line 147: Was this tool developed for this study? How do Q2 and Q10 and Q11 relate to each other, do they indeed ask about different things? Line 247: How does this number relate to the minimum sample size calculated as necessary, when taking into account the provided calculations. Lines 374-376: All three cited references refer explicitly to the situation of the pandemic, while the text of the sentence here aims to speak of the experiences of nursing students and why these lead to higher levels of anxiety, depression, etc. This is not solely due to the pandemic, as this is not a new phenomenon - it has to do with the choice of a responsible helping profession, demanding curriculum, etc. Discussion section should not contain so many citations of results of this study that are already provided within the illustrations in the Results. Discussion needs to include comparisons with what other studies have found on the use of AI tools for mental health support in students. Line 467: The sample size is also a limitation. Conclusion should be more concise and only contain the most important results and their implication, not possible explanations, as this is not a discussion.

Author Response

We sincerely thank you for your thoughtful and constructive review of our manuscript. Your comments have significantly improved the quality and clarity of the work. We have carefully addressed each comment, and the corresponding revisions have been highlighted in red within the revised manuscript. Your insights were instrumental in improving the overall rigor and relevance of the study.

Comment 1: This section needs more focus on the studied population. Also, the references which are cited do not match the text where they are cited - this must be thoroughly checked. For example, on Lines 47-50: The cited references No. 9 and 10 have nothing to do with the text of this sentence - ref. 9 is a preprint (there is an article published) which describes the differences in the "load" of psychiatric consultations before and following the onset of pandemic and ref. No. 10 is a case report of a woman, describing the impact of pandemic on her mental health, while the text of the sentence is about the adoption and effectiveness of AI and telemedicine. This must be corrected and checked throughout the paper.

Response 1: Thank you for pointing this out. We agree with this comment. Therefore , we deleted the sentences concerning COVID-19, as they were not relevant or coherent with the main idea of the research. Additionally, we have thoroughly reviewed and checked all references throughout the manuscript to ensure their relevancy to the text.

Comment 2:  Line 93: "As a result.." - results of what? How does this sentence arise from the entire paragraph?

Response 2 : Thank you for pointing this out. We agree with this comment. Therefore , this sentence was deleted when we restructured the Introduction section. Pages 1-3.

Comment 3:  Lines 97-98: The cited tool refers to which mental health issue experienced by students? It should be stated precisely.

Response 3 : Thank you for pointing this out. We appreciate your comment. Therefore , we have added sentences from the literature to the introduction section to clarify that, while AI-powered mental health tools can support various mental health issues, the literature finds their effectiveness specific for anxiety, depression, and insomnia among students. Page 2, lines 77-81.

Comment 4:  Lines 103-114: This paragraph lacks coherency. It should summarize the main points and outline the aims of the work. For example, what does the text on Line 107 mean - what resources are limited, especially in the educational settings. Line 133: Does any AI-driven tool mean not only related to mental health but any at all?

Response 4 : Thank you for pointing this out. We appreciate your comment. Therefore , we have restructured the last paragraph in the introduction section to make it more coherent and clearly state the aim of the study. The revised paragrapgh can be found on page 3, specifically lines 100-110.

Comment 5: Methods: State explicitly whether approval for the use of questionnaires was received from their owners, or if not required - state so.

Response 5: Thank you for pointing this out. We agree with your comment. Therefore, we have added to the methodology section that the scales are freely available online and were used in their original English form. This information can be found on page 4, line 144.

Comment 6: Methods: . Line 147: Was this tool developed for this study?

Response 6:  Thank you for pointing this out. We appreciate your comment. Yes, this tool was developed specifically for this study, as no suitable existing tools were available in the literature to measure the concepts related to this recent topic.

Comment 7: How do Q2 and Q10 and Q11 relate to each other, do they indeed ask about different things? 

Response 7: Thank you for pointing this out. We appreciate your comment. We would like to clarify that although Q2, Q10, and Q11 all address privacy-related issues, they each focus on different aspects of students’ perceptions. Q2 ("I believe AI-driven mental health applications can maintain confidentiality and protect my personal information.") measures students' confidence in the confidentiality of AI-based mental health applications in general. Q10 ("I am concerned about how my personal data is stored and used when accessing AI-driven mental health services.") extends the general trust and targets students' specific concerns related to the technical processes, storage, and possible abuse of their personal data. Q11 ("I worry that AI-driven mental health applications may share my personal information without my consent.") also focuses on students' concerns about data sharing without their knowledge or consent. Therefore, items Q10 and Q11 were conceived to tap into particular worries about security and privacy, to examine in greater depth issues that go beyond the general idea of general trust reflected in Q2. While students may initially agree that they trust AI-driven mental health tools to maintain confidentiality, it is important to explore whether they have deeper concerns about how their personal data is stored, managed, and potentially used by these tools. 

Comment 8: Line 247: How does this number relate to the minimum sample size calculated as necessary, when taking into account the provided calculations. 

Response 8:  Thank you for pointing this out. We agree with your comment. We acknowledge that the number 180 was written by mistake, likely during the transfer of the manuscript into the journal’s template. We apologize for this oversight. According to the sample size calculation using G*Power (medium effect size f² = 0.15, α = 0.05, power = 0.95, and 9 predictors), the required sample size was calculated to be 166 participants. After applying a 20% increase to account for potential attrition, the final adjusted sample size should correctly be stated as 199 participants. This correction has been made in the revised manuscript to ensure consistency between the methodology and results sections.

Comment 9: Lines 374-376: All three cited references refer explicitly to the situation of the pandemic, while the text of the sentence here aims to speak of the experiences of nursing students and why these lead to higher levels of anxiety, depression, etc. This is not solely due to the pandemic, as this is not a new phenomenon - it has to do with the choice of a responsible helping profession, demanding curriculum, etc.

Response 9: Thank you for pointing this out. We agree with your comment. Therefore, we have revised the section and replaced the previously cited references with more relevant studies independent of the COVID-19 pandemic. The updated references reflect factors such as the demands of nursing education. The updated references can be found on page 12, specifically lines 416-422.

Comment 10:Discussion section should not contain so many citations of results of this study that are already provided within the illustrations in the Results.

Response 10:  Thank you for pointing this out. We agree with your comment. Therefore, we have revised the Discussion section and removed the numerical results, as they were already cited and illustrated in the Results section. Pages 11-13.

Comment 11: Discussion needs to include comparisons with what other studies have found on the use of AI tools for mental health support in students. 

Response 11: Thank you for pointing this out. We agree with your comment. Therefore, we have revised the Discussion section and added recent literature related to the use of AI tools for mental health support among students. However, we also noted that available studies are still limited, as this topic remains relatively new in the field. The study can be found on pages 10-11, specifically lines 379-392.

Comment 12: Line 467: The sample size is also a limitation.Thank you for pointing this out. We agree with your comment. Therefore, we

Response 12: Thank you for pointing this out. We agree with your comment. Therefore, we have added sentences to the Limitations section acknowledging that the sample size is a limitation. This can be found on page 14, specifically lines 519-520.

Comment 13: Conclusion should be more concise and only contain the most important results and their implication, not possible explanations, as this is not a discussion.

Response 13: Thank you for pointing this out. We agree with your comment. Therefore, we have restructured the conclusion section and deleted any explanatory content. The revised conclusion now focuses solely on summarizing the main results and their implications. The revised conclusion can be found on page 14, specifically lines 533-548.

Round 2

Reviewer 2 Report

Comments and Suggestions for Authors I would like to thank the Authors for their careful consideration of the comments and for providing responses to them all. The Introduction and Discussion read much better now, and necessary details were added in the Methods.   There are some minor comments:   - Lines 132-140: Check if this text and Table 1 should actually be in the Results section and not here in the Methods section.  - References 36-37: Add journal and other citation details.

Author Response

We sincerely thank you for your thoughtful and encouraging feedback. We greatly appreciate your recognition of the improvements made across the manuscript. Your constructive comments throughout the review process have been instrumental in strengthening the overall quality and coherence of the work. All modifications made in response to your comments are now clearly highlighted in red.

Comment 1: Lines 132-140: Check if this text and Table 1 should actually be in the Results section and not here in the Methods section.

Response 1: Thank you for pointing this out. We agree with this comment. Therefore, we have relocated the text from Lines 132–140 and Table 1 to the beginning of the Results section.

Comment 2: References 36-37: Add journal and other citation details.

Response 2: Thank you for pointing this out. We agree with this comment. Therefore, we have updated references 36 and 37 to include complete citation details.